# Net cost savings arising from patient completion of an active self-management program

**Maja Gorniak[1], Marvin Pardillo[1]\*, Catherine Keating[2], Courtney Brown[2], Chris Schilling[1]**

**1** KPMG, Economic Modelling, Melbourne, VIC, Australia, **2** Medibank Private, Melbourne, VIC, Australia

\* mgorniak2@kpmg.com.au

**Data Availability Statement:** The data that supports this study were obtained from Medibank Private Ltd and the University of Melbourne by permission. Data will be shared upon reasonable request to the corresponding author with

## Abstract

### Objective

The objective of this study is to investigate changes in willingness for total knee replacement (TKR) surgery following a randomised control trial (RCT) of an osteoarthritis management program, and to extrapolate orthopaedic cost consequences for private health insurers (PHI).

### Methods

Willingness for surgery data from the RCT is analysed using a multinomial logistic regression model. A decision analytic model is used to conduct a break-even cost benefit analysis of the intervention from a PHI payer perspective. The analysis estimates the minimum probability of progression to surgery required for the intervention to be cost-neutral when considering savings limited to reduced orthopaedic costs. Cost data and orthopaedic pathway probabilities are sourced from payer data.

### Results

At baseline, 39% of participants in the treatment and control group were willing for surgery. At 12 months, 16% of participants in the treatment group remained willing for surgery, versus 36% in the control group. Participants in the treatment group are 2.96 (95% CI: 1.01–8.66) times more likely than those in the control group to move from initially willing for surgery, to unsure or unwilling at 12 months. The analysis indicates that the intervention is likely to be cost saving when at least 60% of initially willing participants progress to surgery over a 5-year time horizon.

### Conclusion

Our study estimates that an education, exercise and weight loss intervention can deliver both improved participant outcomes and a return on investment to Australian PHIs through a reduction in TKR surgery incidence.

permission from Medibank Private Ltd and the University of Melbourne. Data from Medibank Private Ltd are available upon request via Kimberly Buck (Kimberly.Buck@medibank.com.au) and Kirsty Miller (Kirsty.Miller@medibank.com.au). Data from the University of Melbourne are available upon request via Kim Bennell (k.bennell@unimelb. edu.au) and Rana Hinman (ranash@unimelb.edu. au).

**Funding:** Medibank Private Limited employees Catherine Keating and Courtney Brown provided assistance with study design and preparation of the manuscript. Maja Gorniak, Marvin Pardillo, and Chris Schilling were employed on a consultancy basis by Medibank Private Limited to design and conduct the economic modelling analysis and prepare the manuscript.

**Competing interests:** I have read the journal's policy and the authors of this manuscript have the following competing interests: Maja Gorniak, Marvin Pardillo and Chris Schilling were employed on a consultancy basis to conduct this economic modelling analysis.

## Introduction

Osteoarthritis (OA) affects approximately 1 in 11 Australians and in 2017–18 OA cost the Australian health system an estimated $3.5 billion, representing 28% of disease expenditure on musculoskeletal conditions and 3% of total disease expenditure [1]. The incidence of total knee replacement (TKR), a treatment for end-stage OA, is estimated to rise by 276% by 2030, increasing real healthcare costs of OA to $AUD5.32 billion, of which $AUD3.54 billion would be borne by the private sector[2]. A 5% reduction in population-level obesity, an important risk factor associated with OA, could result in over 8,000 fewer procedures and savings of approximately $170 million [2, 3].

The most prominent determinant for receiving TKR is a patient's willingness for surgery which is influenced largely by sociodemographic factors and severity of OA symptoms [4–6]. Self-management programs focussing on education, physical activity and weight loss are highly recommended strategies to manage OA, with global studies showing these programs can reduce participant willingness for surgery by between 24% and 67% [7–11].

The aim of this study is to estimate orthopaedic cost savings to Private Health Insurers (PHIs) arising from a reduction in willingness for surgery post participation in an education, exercise and weight loss intervention. Better Knee, Better Me (BKBM) is a remotely delivered OA management program designed to change the path of care for Australians affected by OA, through exercise consultations with a physiotherapist, dietitian consultations and a very low-calorie ketogenic diet [12]. A 12-month randomised controlled trial (RCT) of the program has been administered to eligible participants of an Australian PHI by a consortium of researchers led by the University of Melbourne [13]. The primary outcomes of the study measured changes in knee pain and physical function and have been reported previously [13]. Secondary outcomes included changes in weight and willingness for surgery. This paper provides, for the first time, an analysis of change in willingness for surgery from the RCT and completes an economic evaluation to assess the potential return on investment to PHIs delivering such interventions.

## Methods

Data measuring participants' willingness to undergo surgery were collected from a three-arm RCT involving 415 individuals. All 3 groups received access to online information about OA and self-management. Participants in the Exercise group also received 6 consultations with a physiotherapist over 6 months, strengthening exercise and physical activity program, advice about management, and additional educational resources. The Exercise plus weight management group (treatment) received 6 consultations with a dietitian over 6 months, very low-calorie ketogenic diet with meal replacements and resources to support behaviour change (in addition to all elements of the exercise intervention). The Information only (control) group received access to online information about OA and self-management. Participant data received from the PHI was fully anonymised. Approval was not required as this study provides analysis of results from a previously published RCT [13].

At baseline and at 12 months post program completion, the treatment and control groups were surveyed for their willingness for surgery in the near future. Surveys used a 5-point rating scale with terminal descriptors of 'definitely not willing' to 'definitely willing', with those indicating 'probably willing' or 'definitely willing' classified as willing to have knee surgery. Participants who indicated they were 'unsure' about undergoing knee surgery were grouped separately from participants who indicated a willingness or unwillingness for surgery.

Multinomial logistic regressions were conducted to estimate the probability of shifts in a participant's willingness for surgery status using Stata/SE 17 (StataCorp, TX, USA). Results from the RCT are summarised in Table 1 below.

Due to ethical considerations, long term PHI data on TKRs post program was unavailable as the control group was not maintained. A review of the literature found only 2 studies that examined this relationship [5, 14] and the transferability of these studies were deemed likely to be low due to considerations around participant access to surgery and individual engagement with personal health. Given the lack of progression to surgery data, the probability of surgery given a participant's willingness status was treated as an uncertain parameter. A decision modelling framework that tested all probabilities of surgery given a participant's stated willingness status in the future was used to conduct a break-even cost benefit analysis from the perspective of an Australian PHI. Doing so allowed for an estimation of the minimum probability required for the intervention to be cost-neutral when considering orthopaedic costs.

Given the lack of data on progression to surgery given a participant's willingness status, we also assumed a fixed relationship between the probability of surgery given a participant's willingness status to simplify the analysis. Our model assumed the probability of surgery given an unsure and unwilling status to be 10% and 5% of a willing status, respectively.

Fig 1 below demonstrates the modelling framework for the orthopaedic pathway. The model compares the occurrence and cost of TKR associated with a hypothetical cohort of 1000 participants in the treatment group and control group. The probability of TKR given willingness status was defined as a fixed annual transitional probability over the 5-year time horizon; whilst the number of surgeries per group that occur were analysed in yearly cycles. After each year, a proportion of the cohort undergo surgery while the remainder do not. Participants who do not undergo surgery retain their willingness status and are eligible for surgery in the subsequent 4 years. Any secondary admissions following the initial TKR were also considered in the analysis, up to a further 5 years post initial TKR. Historical PHI data on secondary TJR related hospitalisations between 2016 and 2020 indicated that approximately 3 in 4 members who underwent a TKR required a second TKR, THR, revision or rehabilitation (either at home or in hospital) in the following 5 years.

Orthopaedic pathway probabilities and cost data were derived from the PHI and are summarised in S1 Appendix. Costs associated with TKR, THR, revisions and rehabilitation were estimated using historical cost data from 2016 and 2020. Costs accruing post BKBM were

**Table 1. Summary of survey results.**

| | Treatment group | | | Control group | | |
|---|---|---|---|---|---|---|
| | Mean | Standard error | 95% CI | Mean | Standard error | 95% CI |
| **Baseline** | | | | | | |
| Probably willing | 26% | 0.03 | [0.20, 0.33] | 25% | 0.05 | [0.15, 0.36] |
| Definitely willing | 13% | 0.03 | [0.08, 0.17] | 13% | 0.04 | [0.05, 0.22] |
| Unsure | 35% | 0.04 | [0.28, 0.42] | 33% | 0.06 | [0.22, 0.44] |
| Definitely unwilling | 8% | 0.02 | [0.04, 0.12] | 7% | 0.03 | [0.01, 0.14] |
| Probably unwilling | 18% | 0.03 | [0.13, 0.24] | 21% | 0.05 | [0.11, 0.31] |
| **At 12 months** | | | | | | |
| Definitely willing | 6% | 0.02 | [0.02, 0.10] | 14% | 0.05 | [0.04, 0.24] |
| Probably willing | 10% | 0.02 | [0.05, 0.14] | 22% | 0.06 | [0.11, 0.33] |
| Unsure | 18% | 0.03 | [0.12, 0.24] | 24% | 0.06 | [0.12, 0.36] |
| Probably unwilling | 24% | 0.03 | [0.17, 0.30] | 22% | 0.06 | [0.11, 0.33] |
| Definitely unwilling | 43% | 0.04 | [0.35, 0.50] | 18% | 0.05 | [0.07, 0.29] |

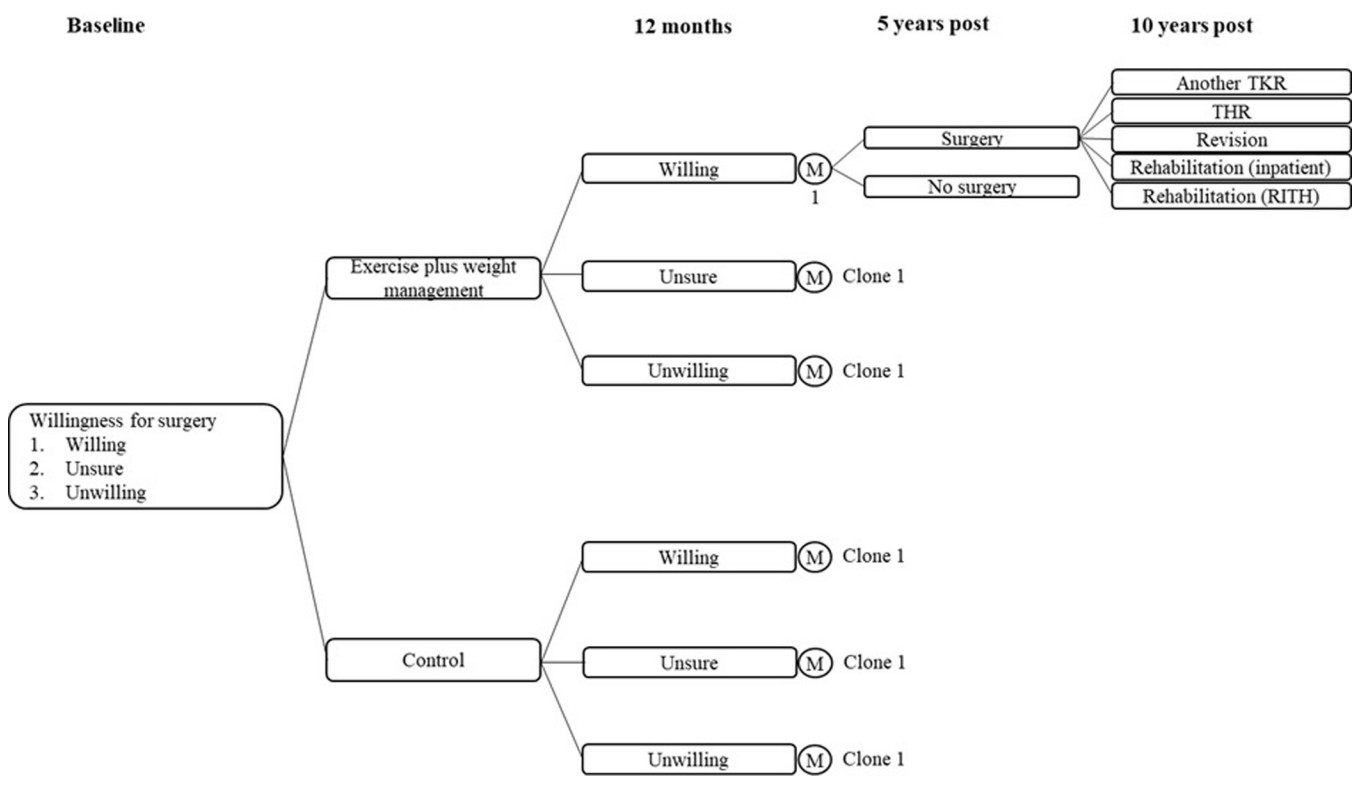

**Fig 1. Decision modelling framework.**

discounted using an annual rate of 3.5% and the growth of costs are modelled as per growth observed between 2016 and 2020. All costs are reported in 2020 Australian dollars. The average net cost saving is reported and calculated by subtracting the cost of TKR surgery between groups and the cost of delivering the program for the treatment group.

A sensitivity analysis was used to examine the uncertainty in model parameter estimates. An example of how the model calculates average net savings for the PHI is outlined in S2 and S3 Appendices.

## Results

Results from this analysis are summarised in Fig 2, which presents the proportion of the treatment and control groups unwilling to have surgery at baseline and at 12 months. Table 2 shows that at baseline, survey outcomes show that in both groups, between 26% and 28% of participants were unwilling for surgery. At 12 months post the program, 66% of participants in the treatment group were unwilling for surgery, versus 40% in the control group. Table 3 indicates that 70% of initially willing participants in the treatment group shift to a less willing status. In comparison, only 44% of initially willing participants in the control group shift to a less willing status. That is, participants in the treatment group are 2.96 (95% CI: 1.01–8.66) times more likely than those in the control group to move to a less willing status.

Table 4 presents the range of average net cost saving versus the probability of surgery given willingness status over a 5-year time horizon. The analysis indicates that average net cost savings increase if we assume a stronger relationship between willingness for surgery and progression to surgery. The break-even analysis indicates that the intervention is cost saving when at least 60% of willing participants go on to have surgery in the following 5 years, returning net

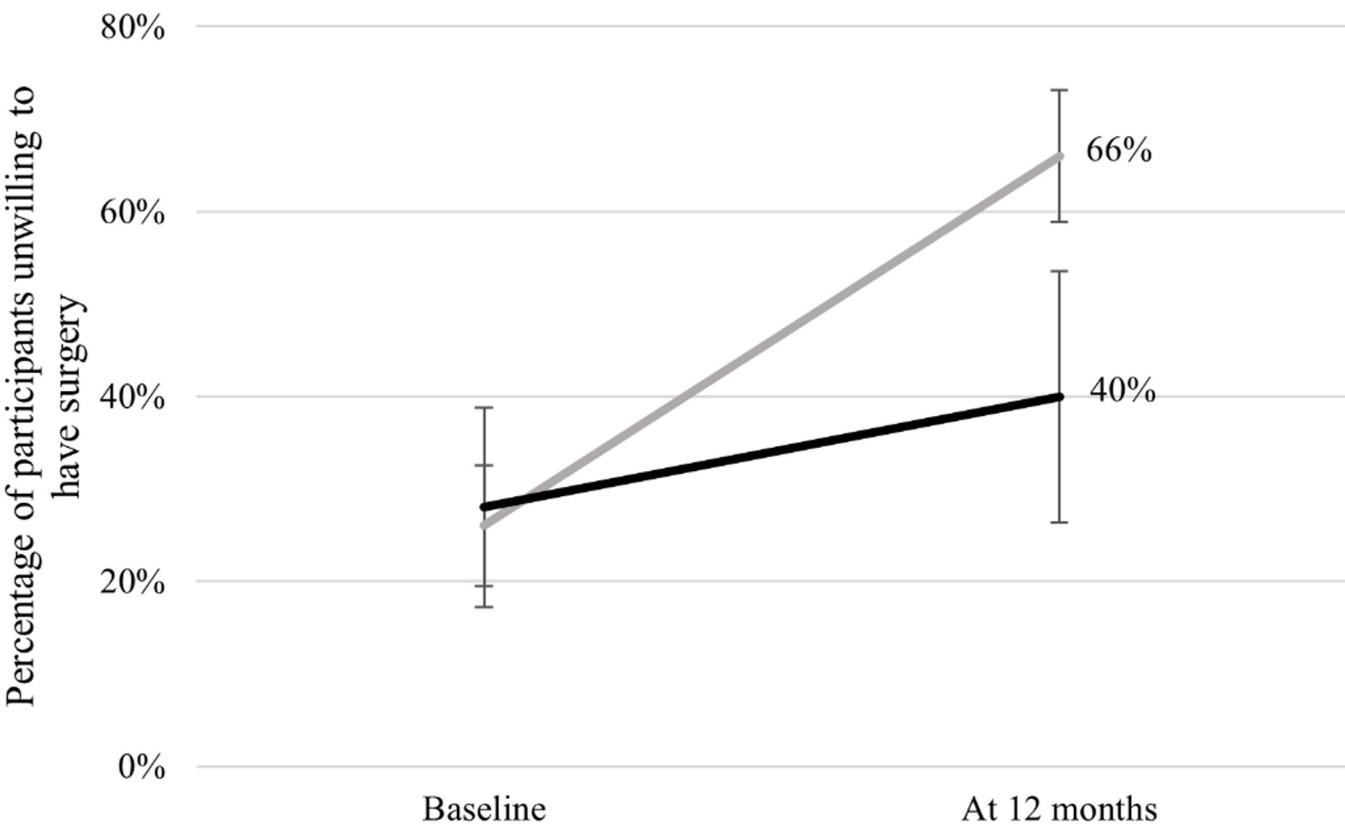

**Fig 2. Program participants are less willing for surgery following program completion.** Note: Error bars indicate the 95% confidence intervals.

savings of approximately $286 per participant. This break-even level can also be converted into an annual probability [15]. After conversion, the annual probability required for the intervention to be cost saving is 5.9%. This result falls in line with the annual rate of knee replacement procedures generally observed among individuals aged over 45, with OA and private health insurance[*] [1, 2].

## Discussion

This study reports on changes to willingness for surgery following an education, exercise and weight loss intervention and extrapolates longer-term cost implications from a PHI perspective. Such interventions are core treatments for knee OA and have been shown to reduce willingness for surgery globally [7–11]. One Australian study reported significant increases in the proportion of participants that become unwilling for surgery after participation in an OA care

**Table 2. Change in willingness status from baseline to 12 months post BKBM participation.**

|  | Treatment group | | Control group | |
|---|---|---|---|---|
|  | Baseline | At 12 months | Baseline | At 12 months |
| Willing | 39% | 16% | 39% | 36% |
| Unsure | 35% | 18% | 33% | 24% |
| Unwilling | 26% | 66% | 28% | 40% |

Note: Please see Table 1 for details on observed survey results prior to categorisation.

**Table 3. Logistic regression analysis.**

| 12 months post BKBM | | Odds ratio (95% CI) | | |
|---|---|---|---|---|
| **Willingness for surgery** | **Treatment** | **Control** | **Treatment vs Control** | **P- value** |
| Remain willing | 19/64 (29.69%) | 10/18 (55.56%) | 1 (ref) | |
| Unsure/unwilling | 45/64 (70.31%) | 8/18 (44.44%) | 2.96 (1.01, 8.66) | 0.05 |

program [7]. Our results are consistent with the findings from these studies; a significant proportion of the treated group were less willing for surgery 12 months post program participation, and this is the main driver of the economic results.

Previous international studies have found similar interventions to be cost-effective or cost-saving when compared to the standard care pathway [16–19]. A recent Australian study modelled potential cost savings associated with implementing a first-line OA management program and found avoidance of TKR could translate to healthcare savings [20]. Results from our orthopaedic cost model align with these studies, indicating that at current rates of surgery, the intervention can deliver significant returns on investment to the PHI.

Our study helps decision makers better understand the economic implications and potential benefits of delivering similar interventions. The model presents evidence to suggest that, when at least 60% of willing participants go on to have surgery in the following 5 years, BKBM may be a dominant intervention: it improves participant health (as demonstrated by the RCT) and delivers a return on investment. Interventions such as BKBM allow PHIs to take an active part in member health and presents a potential win-win for PHIs, members, and the broader health system.

**Table 4. Average discounted net savings for PHI including secondary admissions post initial TKR.**

| Assumed probability of TKR surgery within 5 years given a willing status | Orthopedic surgery cost savings | Average net cost savings ^ | Standard error | Min | Max |
|---|---|---|---|---|---|
| 0 to 0.1 | $137.4 | -$ 2,108.6 | $ 150.7 | -$ 2,246.0 | -$ 1,673.5 |
| 0.11 to 0.2 | $467.1 | -$ 1,778.9 | $ 145.5 | -$ 2,046.1 | -$ 1,451.4 |
| 0.21 to 0.3 | $640.2 | -$ 1,605.8 | $ 242.9 | -$ 2,044.6 | -$ 1,007.6 |
| 0.31 to 0.4 | $1,158.1 | -$ 1,087.9 | $ 366.0 | -$ 1,953.7 | -$ 97.4 |
| 0.41 to 0.5 | $1,557.6 | -$ 688.4 | $ 552.6 | -$ 1,830.3 | $ 335.8 |
| 0.51 to 0.6 | $1,941.6 | -$ 304.4 | $ 643.2 | -$ 1,742.4 | $ 1,127.5 |
| 0.61 to 0.7 | $2,532.3 | $ 286.3 | $ 774.1 | -$ 2,145.9 | $ 1,742.8 |
| 0.71 to 0.8 | $3,067.0 | $ 821.0 | $ 1,019.3 | -$ 1,756.2 | $ 3,812.5 |
| 0.81 to 0.9 | $3,492.9 | $ 1,246.9 | $ 1,143.0 | -$ 1,157.3 | $ 5,822.3 |
| 0.91 to 1.0 | $4,895.8 | $ 2,649.8 | $ 1,675.0 | -$ 1,609.4 | $ 9,624.4 |

* Osteoarthritis and TKR data from the Australian Institute of Health and Welfare (AIHW) is used to calculate the annual rate of knee replacement procedures. The result has been adjusted to account for the share of TKR surgeries that occur in the private sector.

There are important limitations to our study. First, due to the unavailability of long-term surgical data from the PHI, our study explored orthopaedic cost consequences under a range of possible probability levels for surgery given a participant's willingness status. To address the uncertainty in this parameter estimate, we have conducted a break-even analysis to test the variability of our results. The unavailability of long-term surgical PHI data also meant our model assumed a fixed relationship between willingness states and progression to surgery.

Second, the economic evaluation considered a narrow set of benefits relating to reduced TKR healthcare utilisation; there are a range of further benefits to participants and payers that have not been considered, such as the impact of weight loss, improved health and lower non-TKR healthcare utilization [21]. A recent microsimulation model of OA in Canada found that a 1-unit change in body mass index can lead to a substantial reduction in disability in the long term [21].

Third, results presented in this study do not account for possibility that participants become more likely to undergo surgery after achieving weight loss. Obesity is a significant predictor of surgical failures, making it a concern to most surgeons [22, 23]. Less willing participants who lose weight in the trial may become more attractive to surgeons as candidates for surgery, in which case participant outcomes improve but costs to the PHI increase. Additional research to better understand the occurrence of surgery given a participant's willingness status and the possibility of perverse outcomes would assist with validating the robustness of extrapolating trial results to economic impact.

Fourth, our study assumed a fixed transition probability for TKR over the 5-year study period. Research into the progression to surgery suggests there are multiple pathways, some faster and some slower, so on balance we believe this is a reasonable assumption [24]. However, given cost savings are discounted, a higher likelihood of surgery in the early years relative to the later years would increase the pay-off to the intervention.

Finally, by adopting a PHI payer perspective, the transferability of our findings in public sector settings may be low. Participants in our study had access to surgery and were more likely to be engaged with their personal health.

## Conclusion

The results of this study have shed light on orthopaedic cost savings to Australian PHIs arising from a reduction in willingness for TKR surgery following an education, exercise and weight loss intervention. Results from our decision modelling framework indicate that at current rates of surgery such interventions can deliver significant returns on investment to the PHI. The delivery of interventions such as BKBM allow PHIs to take an active part in improving member health which may produce monetary benefits in the form of reduced total benefit outlay. However, additional research into the relationship between willingness status and progression to surgery is needed to assist with validating the robustness of extrapolating trial results to economic impact.

## Supporting information

**S1 Appendix. Table 4. Model parameters.**
(DOCX)

**S2 Appendix. Table 5. Cost and surgical rates for initial TKR assuming a probability rate of 0.65.**
(DOCX)

**S3 Appendix. Table 6. Cost and surgical rates for initial TKR assuming a probability rate of 0.65.**
(DOCX)

## Acknowledgments

We acknowledge the contributions of Medibank Private Ltd and the University of Melbourne for their contribution to this study.

## Author Contributions

**Conceptualization:** Maja Gorniak, Marvin Pardillo.

**Data curation:** Maja Gorniak, Marvin Pardillo.

**Formal analysis:** Maja Gorniak, Marvin Pardillo.

**Investigation:** Maja Gorniak, Marvin Pardillo.

**Methodology:** Maja Gorniak, Marvin Pardillo.

**Project administration:** Maja Gorniak, Marvin Pardillo.

**Resources:** Maja Gorniak.

**Software:** Maja Gorniak.

**Supervision:** Maja Gorniak.

**Visualization:** Marvin Pardillo.

**Writing – original draft:** Marvin Pardillo.

**Writing – review & editing:** Maja Gorniak, Marvin Pardillo, Catherine Keating, Courtney Brown, Chris Schilling.

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
