## [Decision Letter · Decision Letter 0]

20 Jan 2023

PONE-D-22-29718Net cost savings arising from patient completion of an active self-management program

PLOS ONE

Dear Dr. Marvin Pardillo,

Thank you for submitting your manuscript to PLOS ONE. After careful consideration, we feel that it has merit but does not fully meet PLOS ONE’s publication criteria as it currently stands. Therefore, we invite you to submit a revised version of the manuscript that addresses the points raised during the review process.

The manuscript has been evaluated by one reviewer, and their comments are available below. The reviewer have raised a number of concerns that need attention. They found the introduction unclear and request additional information on methodological aspects of the study. Could you please revise the manuscript to carefully address the concerns raised?

Please note that we have only been able to secure a single reviewer to assess your manuscript. We are issuing a decision on your manuscript at this point to prevent further delays in the evaluation of your manuscript. Please be aware that the editor who handles your revised manuscript might find it necessary to invite additional reviewers to assess this work once the revised manuscript is submitted. However, we will aim to proceed on the basis of this single review if possible. 

We look forward to receiving your revised manuscript.

Kind regards,

Katrien Janin, PhD

Staff Editor

PLOS ONE

Journal Requirements:

Reviewers' comments:

Reviewer's Responses to Questions

**Comments to the Author**

1. Is the manuscript technically sound, and do the data support the conclusions?

Reviewer #1: Partly

2. Has the statistical analysis been performed appropriately and rigorously? 

Reviewer #1: I Don't Know

3. Have the authors made all data underlying the findings in their manuscript fully available?

Reviewer #1: Yes

4. Is the manuscript presented in an intelligible fashion and written in standard English?

Reviewer #1: Yes

5. Review Comments to the Author

Reviewer #1: The issues with this study are to do with the mix of input data and assumptions:

1) it begins with the premise that it will be a quantitative economic evaluation of observed data from an RCT, and thus have the rigour of causal inference on its side, but...;

2) it cannot due to a key unknown: the prior probability of a person progressing to surgery (over a given period) given the person's willingness status.

This means that the problem has both: 1) parameter uncertainty from observation error (i.e. the observed values of willingness are estimates from a sample of an underlying population, and thus are uncertain) as well as 2) assumption uncertainty (from the key unknown - assumed probability - forcing the analysts to test a range of plausible parameters).

And this shift from the initial premise to what was actually done makes the abstract very unclear for the reader. Expectations of a quantitative economic evaluation of observed data from an RCT are suddenly undermined in the space of a couple of sentences when suddenly "the analysis estimates the minimum probability required for the intervention to be..." - probability of what?? The reader has no idea yet what probability you're talking about. Nowhere in the abstract does this mysterious probability get revealed. The results report a proportion (60%) of initially willing participants proceeding to surgery. The study does not answer what proportion of initially willing participants actually did proceed to surgery, because it couldn't be observed in the trial beyond 1 year (or at all?) - has that actual number been reported anywhere?

The latter (2) makes this a Bayesian problem, in which the prior (2) has a strong influence on the results, with uncertainty (1) providing an uncertainty interval around all of the assumed probability input values.

The methods do seem to capture all of this, but not very clearly. For example it is stated that "The probability

of surgery given a participant's willingness status was treated as an uncertain parameter" (line 91) - I recommend you state up front in this paragraph that you were forced to make assumptions about this key unknown, and due to lack of good evidence were forced to model the full range of potential assumed values (from 0 to 1.0).

In the Discussion section, you list this as the second limitation - it should feature much more prominently in the Discussion than that.

Then, be sure to distinguish this primary assumption from the secondary assumption (lines 100-3).

I'm not clear where the standard errors in Table 3 come from. Are they from sampling uncertainty on the Likert scale of willingness? Table 5 makes me think that it is. I think Table 5 needs to come much earlier.

I think part of the lack of clarity is from the order we are provided the key information in. Also, it would be great to see a Figure or two with nice data visualisation, for example Table 1 could be presented visually showing what share of people shift from 'willing' to 'unsure' or 'unwilling', etc, over time.

Table 5 is not consistent with Table 1: At baseline, Table 1 says that 39% of participants in both groups were "willing", but Table 5 says it was only 26% in the treatment group and 28% in the control group. This corresponds to the share that were "unwilling" in Table 1, so the classification labels may be transposed. Which is correct? This is a critical error, and could presumably make a big difference to the results.

Table 3 shows that the estimate for net savings becomes positive (cost saving) when the assumed probability of TKR surgery, given willingness, is 0.61 to 0.7. However the uncertainty interval based on the standard errors never excludes 0 even at the highest probability bracket. This is not entirely consistent with your Results and Conclusion in the abstract that claiming that "the intervention is cost saving", in terms of the uncertainty interval.

I recommend stating very clearly in the abstract that the analysis is limited to the very narrow set of benefits (from lines 166-8).

The Tables from the main paper seem to be repeated unnecessarily in the Appendices.

Please address the assumption of constancy of the assumed probability over time (line 109) in the discussion.

The interpretation is also too vague to make the study easily understandable and interpretable. In several place you state that "under certain conditions" and "in some circumstances..." (lines 159, 162, 187) the intervention may be "a dominant intervention", but don't clearly state what conditions, what circumstances? This should be made much clearer.

The main driver appears to be (difference in) share of people who shift from willing or unsure to unwilling (according to Table 1, at least - but given inconsistency with Table 5 it's confusing). I suggest making this clearer in the discussion.

The other main driver of course is the unknown key assumption.

Give some clear guidance to help future researchers be very clear about what gaps in knowledge need to be addressed.

6. PLOS authors have the option to publish the peer review history of their article (what does this mean?). If published, this will include your full peer review and any attached files.

Reviewer #1: No

---

## [Author Response · Author response to Decision Letter 0]

4 Jun 2023

Thank you for your comments and suggestions on our manuscript. We have provided responses to each of your points in the attached Response to Reviewers document as part of our re-submission.

---

## [Decision Letter · Decision Letter 1]

26 Jun 2023

PONE-D-22-29718R1Net cost savings arising from patient completion of an active self-management programPLOS ONE

Dear Dr. Pardillo,

Thank you for submitting your manuscript to PLOS ONE. After careful consideration, we feel that it has merit but does not fully meet PLOS ONE’s publication criteria as it currently stands. Therefore, we invite you to submit a revised version of the manuscript that addresses the points raised during the review process.

We look forward to receiving your revised manuscript.

Kind regards,

Filippo Migliorini MD, PhD, MBA

Academic Editor

PLOS ONE

Journal Requirements:

Additional Editor Comments:

I ask the authors if they could suggest a couple of reviewers for their submission. I need at least two reviewers to expedite the revision and accept their work. Thank you. FM

Reviewers' comments:

Reviewer's Responses to Questions

**Comments to the Author**

1. If the authors have adequately addressed your comments raised in a previous round of review and you feel that this manuscript is now acceptable for publication, you may indicate that here to bypass the “Comments to the Author” section, enter your conflict of interest statement in the “Confidential to Editor” section, and submit your "Accept" recommendation.

Reviewer #1: (No Response)

2. Is the manuscript technically sound, and do the data support the conclusions?

Reviewer #1: Yes

3. Has the statistical analysis been performed appropriately and rigorously? 

Reviewer #1: Yes

4. Have the authors made all data underlying the findings in their manuscript fully available?

Reviewer #1: No

5. Is the manuscript presented in an intelligible fashion and written in standard English?

Reviewer #1: Yes

6. Review Comments to the Author

Reviewer #1: Overall the framework is a sound one for answering this question, and the lack of observed surgery data is now clearer and well enough explained.

Please specifically say whether it's a microsimulation model or what...

The sensitivity analyses use an ok approach.

Much clearer now with the willingness status data in the tables and the modelling framework figure.

Clarify why only 1000 in the hypothetical cohort through the model (or is it 1000 in each of the intervention and control groups - it's ambiguous).

Line 149-150: an annual probability of 5.9% would result in only 27% conversion, not 60% - which would change the conclusions - please explain.

Table 4: the left column gives a range, but the other columns seem to come from a single value - clarify whether that single value the mid-point of the range, or are the values the average of model runs across all values in the range, or what specifically.

- clarify whether the standard error and min-max range are due to parameter uncertainty or sampling variation between cohort runs, or both, or from sensitivity analyses.

The discussion states (line 166) that willingness "is the main driver" - may be overstating it as it was the only driver considered in this study.

In the discussion, acknowledge the following as limitations:

- assumption of the probability of surgery for unsure and unwilling to be 10% and 5% was fixed, and not varied in sensitivity analyses.

Appendix - table 4 line 312: why no uncertainty in the lower half of the table? these seem to be estimated from data.

Appendix - table 4 line 313: These groups should not really differ at baseline in the model. Given people are randomly allocated to group, these are chance error only. The model should start both groups the same.

7. PLOS authors have the option to publish the peer review history of their article (what does this mean?). If published, this will include your full peer review and any attached files.

Reviewer #1: No

---

## [Author Response · Author response to Decision Letter 1]

28 Sep 2023

Dear Editor,

Thank you for the opportunity to clarify our competing interests. Our responses to your questions are listed below:

1. Please clarify the sources of funding (financial or material support) for your study. List the grants or organizations that supported your study, including funding received from your institution.

Medibank Private provided funding for the health economists to complete the economic analysis conducted in this study. 

2. State what role the funders took in the study. If the funders had no role in your study, please state: “The funders had no role in study design, data collection and analysis, decision to publish, or preparation of the manuscript.”

Medibank Private employees Catherine Keating and Courtney Brown provided assistance with study design and preparation of the manuscript. 

3. If any authors received a salary from any of your funders, please state which authors and which funders.

Maja Gorniak, Marvin Pardillo, and Chris Schilling were employed on a consultancy basis by Medibank Private to design and conduct the economic modelling analysis, and prepare the manuscript.

---

## [Editor Report · Decision Letter 2]

11 Oct 2023

Net cost savings arising from patient completion of an active self-management program

PONE-D-22-29718R2

Dear Dr. Pardillo,

We’re pleased to inform you that your manuscript has been judged scientifically suitable for publication and will be formally accepted for publication once it meets all outstanding technical requirements.

Kind regards,

Filippo Migliorini MD, PhD, MBA

Academic Editor

PLOS ONE

Additional Editor Comments (optional):

Well done
---

## [Editor Report · Acceptance letter]

18 Oct 2023

PONE-D-22-29718R2 

Net cost savings arising from patient completion of an active self-management program 

Dear Dr. Pardillo:

I'm pleased to inform you that your manuscript has been deemed suitable for publication in PLOS ONE. Congratulations! Your manuscript is now with our production department. 

Kind regards, 

on behalf of

Dr Filippo Migliorini 

Academic Editor

PLOS ONE